# Does use of GP and specialist services vary across areas and according to individual socioeconomic position? A multilevel analysis using linked data in Australia

Danielle C Butler [ORCID],[1] Sarah Larkins [ORCID],[2] Louisa Jorm,[3] Rosemary J Korda[1]

¹National Centre for Epidemiology and Population Health, Australian National University, Canberra, Australian Capital Territory, Australia
²College of Medicine and Dentistry, James Cook University, Townsville, Queensland, Australia
³Centre for Big Data Research in Health, University of New South Wales, Sydney, New South Wales, Australia

**Correspondence to**
Dr Danielle C Butler;
Danielle.Butler@anu.edu.au

## ABSTRACT

**Objective** Timely access to primary care and supporting specialist care relative to need is essential for health equity. However, use of services can vary according to an individual's socioeconomic circumstances or where they live. This study aimed to quantify individual socioeconomic variation in general practitioner (GP) and specialist use in New South Wales (NSW), accounting for area-level variation in use.

**Design** Outcomes were GP use and quality-of-care and specialist use. Multilevel logistic regression was used to estimate: (1) median ORs (MORs) to quantify small area variation in outcomes, which gives the median increased risk of moving to an area of higher risk of an outcome, and (2) ORs to quantify associations between outcomes and individual education level, our main exposure variable. Analyses were adjusted for individual sociodemographic and health characteristics and performed separately by remoteness categories.

**Setting** Baseline data (2006–2009) from the 45 and Up Study, NSW, Australia, linked to Medicare Benefits Schedule and death data (to December 2012).

**Participants** 267 153 adults aged 45 years and older.

**Results** GP (MOR=1.32–1.35) and specialist use (1.16–1.18) varied between areas, accounting for individual characteristics. For a given level of need and accounting for area variation, low education-level individuals were more likely to be frequent users of GP services (no school certificate vs university, OR=1.63–1.91, depending on remoteness category) and have continuity of care (OR=1.14–1.24), but were less likely to see a specialist (OR=0.85–0.95).

**Conclusion** GP and specialist use varied across small areas in NSW, independent of individual characteristics. Use of GP care was equitable, but specialist care was not. Failure to address inequitable specialist use may undermine equity gains within the primary care system. Policies should also focus on local variation.

## INTRODUCTION

Adequate and timely access to primary care relative to need is a specified goal of high-performing health systems.[1–3] This is integral to improving average levels of population health, as well as health equity.[4] Further, an effective primary care system requires ready access to supporting specialist care. Yet, often individuals' socioeconomic circumstances or where they live, as much as their need for care, determine their use of services[5–8]; that is, access to care is inequitable. Examining and quantifying these differing sources of variation in care is essential for directing policy responses for achieving an equitable healthcare system.

There is evidence internationally,[9 10] and to a lesser extent within Australia,[7 8 11–13] of socioeconomic variation in use of general practitioner (GP) and specialist services. Across most jurisdictions, people who are of low socioeconomic position (SEP) use equal or more GP services for a given level of need relative to their high-SEP counterparts.[7–11 13] On the other hand, individuals of high SEP are more likely to see a specialist than those of low SEP.[8–13] Use of primary care and specialist services also varies geographically. Studies in Australia using aggregated area-level data consistently find increased use of GP and specialist services in major cities compared

with more remote areas.[5 12 14] To date, no Australian studies have examined individual socioeconomic variation in use of primary and specialist services while accounting for area variation in use of services or quantified the extent of variation at the area level, beyond that explained by the characteristics (such as sociodemographics or health status) of individuals living in those areas.

The aim of this study was to use large-scale linked data and multilevel analysis[15 16] to examine the extent to which GP and specialist service use varied at the area level, having accounted for the sociodemographic and health characteristics of people who lived in those areas. Further, we quantified variation in use of services according to individual SEP, having accounted for variation in use across areas. In this way, sources of variation in use of GP and specialist services are clarified and indicate directions for reducing unwarranted variation in care.

## METHODS
### Study population and setting
The Sax Institute's 45 and Up Study is a large prospective cohort study involving 267 153 people aged 45 years and older residing in New South Wales (NSW), the most populous state in Australia.[17] Participants were randomly sampled from the Services Australia (formerly the Australian Government Department of Human Services) Medicare enrolment database, with oversampling by a factor of two of individuals aged 80 years and over and people resident in rural areas. Participants enrolled in the study by completing a baseline questionnaire, distributed between 2006 and 2009, and providing consent for five yearly questionnaires and linkage to routinely collected health data. About 19% of those invited participated in the study and participants included ~11% of the total NSW population aged 45 years and older.[18] The study design and details of the questionnaire are reported elsewhere.[18]

### Data
Sociodemographic and health variables were derived from the self-reported baseline questionnaire. Data from the questionnaire were linked to Medicare Benefits Schedule (MBS) claims data (1 January 2003 to 14 December 2012) provided by Services Australia, and data from the NSW Registry of Births, Deaths and Marriages (RBDM) and the National Death Index. The MBS claims database includes all claims for subsidised medical and diagnostic services provided by registered medical and other practitioners through the MBS. For each claim for service processed, the MBS data include a range of information, including the date of the service and the item number for the service. Linkage of baseline data from 45 and Up Study participants to MBS data was performed at the Sax Institute through deterministic linkage, using an encrypted version of the Medicare number provided directly by Services Australia.

Probabilistic linkage to NSW RBDM was performed by the Centre for Health Record Linkage (CHeReL) data.

Quality assurance data on the CHeReL data linkage show false positive and negative rates of <0.5% and <0.1%, respectively.[19]

### Variables
#### Outcomes of interest
For use of GP services, the main outcome was above-average GP use (no/yes) as a measure of frequent use, defined as eight or more services in the year following completion of the baseline survey, which is broadly consistent with definitions reported in the literature.[8 20] We also examined secondary outcomes relating to types and qualities of GP services that indicate high-quality primary care,[1–3] and that the general population would be eligible to receive. This included: (1) Any MBS service for a long or prolonged consultation (no/yes) in the follow-up period (known to be associated with more problems managed and better outcomes[21 22]). (2) Continuity of GP care measured by the usual provider continuity index (UPI),[23] calculated as the proportion of GP MBS services with the most frequent provider of total GP MBS services and defined as a UPI of 70% or more. As per standard definitions, the UPI was calculated over a 2-year period and calculated only for those participants who used at least four GP services in that time. (3) Care planning (no/yes) defined as at least one MBS service for a chronic disease and complex care planning item (including a GP management plan, team care arrangement, or review item) in the follow-up period. These items relate to specific MBS-funded services that can be claimed for care planning relating to chronic and complex care needs and to enable multidisciplinary coordination of care.

Specialist use was defined as any out-of-hospital MBS specialist service in the follow-up period (no/yes). See online supplemental tables 1 and 2 for full list of MBS item codes included in the outcome measures.

#### Main exposure variable
Individual-level sociodemographic and health characteristics were derived from the 45 and Up baseline questionnaire. Our main exposure variable, SEP, was measured as the highest education level attained (no school certificate, school certificate, apprenticeship or diploma, and university degree).

#### Determinants of healthcare need and confounders
SEP is associated with morbidity and health status,[24 25] which in turn determines the need for healthcare. To determine need-adjusted use, healthcare need[8 26] variables included were: self-reported health (excellent, very good, good, fair, and poor); physical functioning (no limitation, minor limitation, moderate limitation, severe limitation, and a missing category); and number of chronic conditions for the following self-reported conditions—cancer, asthma, hay fever, heart disease, heart attack, angina, stroke, diabetes, hypertension, hypercholesterolaemia, arthritis, osteoporosis, anxiety, depression,

and Parkinson's disease (none, 1–2 chronic conditions, 3 or more chronic conditions).

To account for confounders in the relationship between SEP or healthcare need and use of health services we also included: age (10 age categories from 45 years through to 85 years and over); sex (male/female); country of birth (Australia/New Zealand, Europe/North America, Asia, Africa/Middle East, and other); and marital status (married/de facto or not married/not de facto).

Using Australian Bureau of Statistics concordance files, each participant was assigned to a Statistical Area Level 3 (SA3) geography. These areas have populations of between 30 000 and 130 000 persons and are considered representative of communities sharing similar characteristics in terms of services available.

## Analysis

Participants were followed for 1 year after completing the baseline survey (most had completed this by 2008) and for 2 years for continuity of care. We included participants if they had at least one Medicare record, were alive at the end of the follow-up period, had a geographical identifier coded to NSW, had at least four visits in the follow-up period (for analyses of continuity of care), and had at least one long-term health condition (for analyses of care planning).

Frequencies and proportions were calculated for the sample according to participant characteristics and outcomes, for the total sample and by education. A two-level random intercept multilevel logistic regression model (participants nested within SA3 of residence) was fitted for each outcome. To determine if there was significant area-level variation in outcomes, and hence a random intercept model appropriate, a model without explanatory variables (null model) was first fitted (online supplemental tables 3–5). For the main analysis, the model was adjusted for individual education, healthcare need and confounders to determine need-adjusted individual-level socioeconomic variation in outcomes (having accounted for area-level variation).

Area-level variation in each outcome was estimated from the variance term $(V_A)$ by calculating the intraclass correlation coefficient (ICC) by the linear threshold model method (ICC=$V_A/(V_A+3.29)$) and the median OR (MOR=$\exp\left(0.954\sqrt{V_A}\right)$).[15] Conceptually, the MOR is the median of the ORs calculated from pairwise comparison of people with identical covariates, but residing in different areas. It quantifies the median increased risk that would occur if moving from one area to another with higher risk.[15] The proportional change in variance (PCV=$(V_A-V_B/V_A)\times100$)[15] was used to estimate the proportion of overall variation in outcomes explained by the addition of explanatory variables to the model. Second-order penalised quasilikelihood estimation was used as per Rasbash and colleagues.[16] Markov chain Monte Carlo estimation was used to assess model fit statistics and residuals plotted to test model assumptions held.

As health service use in Australia varies according to remoteness, analyses were stratified by categories of remoteness (major cities, inner regional, outer regional/remote) based on the 2006 Access and Remoteness Index of Australia (+)[27] and according to the Australian Statistical Geography Standard Classification of Remoteness Area.

Analyses were undertaken using Stata (V.14.1, StataCorp, College Station, Texas) in the Secure Unified Research Environment, a secure remote access computer facility for analysis of linked data. Multilevel analysis was performed using the runmlwin add-on,[28] using Stata's postestimation procedures.

Sensitivity analyses were also repeated using alternative measures of frequency of GP use (low vs medium and medium vs high) and including those who died in the follow-up period.

## Patient and public involvement

No patients or members of the public were involved in the design or conduct of the study.

## RESULTS

### Sample characteristics

After excluding those who had an invalid death date or died in the follow-up period (n=320), did not have an MBS service (n=1583) or were unable to be assigned to an SA3 (n=151), the final sample for inclusion was 263 083. Of these, 11.7% had no school certificate, 31.8% completed a school certificate, 31.8% had completed an apprenticeship or diploma, and 23% had completed a tertiary-level qualification. The mean age of the population was 62.7 years (SD 11.2), 46% were male, over 80% rated their health as good/very good/excellent and 73% had at least one chronic condition (table 1).

### Area-level variation

Use of GP services varied according to where a person resided—for all regions—having accounted for the characteristics of individuals living in those areas (figure 1, MOR major cities 1.34, inner regional 1.32, outer regional/remote 1.35). This means that an individual who lived in an area with a higher rate of above-average GP use had a (median) 32–35% greater probability of having above-average GP use than an individual with identical characteristics who lived in an area with a lower rate of above-average GP use. Quality of GP care similarly varied across areas (online supplemental table 4). Area-level variation in specialist use across all regions was also evident after accounting for the characteristics of individuals (MOR 1.16–1.18; online supplemental table 5).

### Individual-level socioeconomic variation

People of low education level used more GP services on average compared with those with higher levels of education, for a given level of need and having accounted for area variation in use (figure 2). For secondary outcomes

**Table 1** Sample characteristics: individual-level variables by educational attainment (%) and for total sample

| Variable | Educational attainment | | | | | Row category total % (n) |
|---|---|---|---|---|---|---|
| | No school certificate | School certificate | Apprentice/ diploma | University | Missing | |
| Education | | | | | | |
| Total % (n) | 11.7 (31 126) | 31.8 (84 302) | 31.8 (84 294) | 23 (60 933) | 1.7 (4428) | 100 (265 083) |
| Sex | | | | | | |
| Male | 42.1 | 36.1 | 55.2 | 50.0 | 48.6 | 46 (122 893) |
| Female | 57.3 | 63.9 | 44.8 | 50.0 | 51.5 | 53.6 (142 190) |
| Age | | | | | | |
| 45–54 | 16.5 | 24.4 | 31.6 | 40.1 | 14.4 | 29.2 (77 397) |
| 55–64 | 27.4 | 32.9 | 31.9 | 34.8 | 22.6 | 32.2 (85 342) |
| 65–74 | 28.9 | 23.7 | 21.5 | 15.7 | 25.4 | 21.8 (57 734) |
| 75–84 | 21.8 | 15.3 | 12.7 | 8.0 | 28.8 | 13.8 (36 516) |
| 85+ | 5.4 | 3.8 | 2.3 | 1.5 | 8.8 | 3.1 (8082) |
| Country of birth | | | | | | |
| Australia/New Zealand | 75.9 | 80.8 | 76.9 | 72.5 | 64.8 | 76.8 (203 629) |
| Europe/North America | 18.6 | 13.4 | 17.7 | 16.8 | 20.9 | 16.3 (43 154) |
| Asia | 2.3 | 2.6 | 2.3 | 6.5 | 4.0 | 3.4 (9031) |
| Africa/Middle East | 1.1 | 1.5 | 1.3 | 2.7 | 1.7 | 1.7 (4397) |
| Other | 0.5 | 0.8 | 0.9 | 0.8 | 0.9 | 0.08 (2145) |
| Marital status | | | | | | |
| Not married/not de facto | 32.6 | 26.2 | 22.3 | 21.0 | 33.5 | 24.7 (65 288) |
| Married/de facto | 66.8 | 73.3 | 77.1 | 78.5 | 64.3 | 74.7 (198 185) |
| Self-rated health | | | | | | |
| Excellent | 7.4 | 12.2 | 14.1 | 22.4 | 10.1 | 14.6 (38 575) |
| Very good | 25.6 | 35.0 | 36.8 | 40.8 | 24.6 | 35.6 (94 481) |
| Good | 36.3 | 34.5 | 33.8 | 26.5 | 32.1 | 32.6 (86 451) |
| Fair | 20.5 | 12.3 | 10.7 | 6.9 | 17.5 | 11.6 (30 644) |
| Poor | 4.8 | 2.2 | 1.8 | 1.0 | 4.0 | 2.1 (5575) |
| Chronic conditions | | | | | | |
| None | 20.7 | 25.4 | 27.3 | 31.2 | 25.2 | 26.8 (70 991) |
| 1–2 | 49.9 | 52.0 | 52.6 | 53.0 | 50.1 | 52.1 (198 116) |
| 3 or more | 29.4 | 22.6 | 20.2 | 15.8 | 24.7 | 21.1 (55 976) |
| Physical functioning | | | | | | |
| No limitation | 18.9 | 26.5 | 30.0 | 39.4 | 19.2 | 29.5 (78 323) |
| Minor limitation | 15.4 | 23.2 | 26.9 | 30.2 | 14.5 | 24.9 (66 072) |
| Moderate limitation | 21.4 | 22.5 | 21.6 | 17.3 | 16.7 | 20.8 (55 097) |
| Severe limitation | 21.8 | 12.9 | 10.3 | 5.5 | 16.5 | 11.5 (30 367) |
| GP use | | | | | | |
| Below average | 46.5 | 59.0 | 64.4 | 74.3 | 47.8 | 62.6 (165 803) |
| Above average | 53.6 | 41.1 | 35.6 | 25.7 | 52.2 | 37.5 (99 280) |
| Continuity of care | | | | | | |
| <70% | 41.1 | 44.8 | 46.5 | 50.4 | 41.6 | 46.1 (105 433) |
| ≥70% | 58.7 | 55.0 | 53.4 | 49.6 | 58.1 | 53.7 (128 055) |
| Any care planning | | | | | | |
| No | 53.5 | 54.8 | 55.3 | 56.3 | 51.6 | 55.1 (146 046) |

Continued

**Table 1** Continued

| Variable | Educational attainment | | | | | |
| --- | --- | --- | --- | --- | --- | --- |
| | No school certificate | School certificate | Apprentice/ diploma | University | Missing | Row category total % (n) |
| Yes | 22.4 | 15.9 | 13.5 | 8.5 | 20.1 | 14.3 (37 815) |
| Any long consult | | | | | | |
| No | 56.5 | 58.7 | 60.1 | 59.8 | 56.2 | 59.1 (156 652) |
| Yes | 43.5 | 41.3 | 39.9 | 40.2 | 43.8 | 40.9 (108 428) |
| Any specialist use | | | | | | |
| No | 40.5 | 44.3 | 46.5 | 47.9 | 40.5 | 45.3 (120 063) |
| Yes | 59.5 | 55.7 | 53.6 | 52.1 | 59.5 | 54.7 (145 019) |

Columns for each variable category for each educational attainment category sum to 100%. Values in last column give breakdown by category for each individual variable for the total sample, not stratified by educational attainment. For each variable, total (n) sums to 265 083 and per cent sums to 100% including missing data; $\chi^2$ test for trend with education, p<0.001 all variables. Missing: age <1%, country of birth 1%, marital status 0.6%, self-rated health 3.5%, physical functioning 13.3%. For analyses of continuity of care, those who died in the second year or had less than four GP MBS services during follow-up were excluded (n=31 595). Additional exclusions included: (1) care planning participants without a chronic disease (n=81 222) and (2) outlier observation of long consultations (n=3).
GP, general practitioner.

examining quality of GP care, people of low education level were also more likely to have care planning (eg, no school certificate vs university educated in major cities 1.53 (1.42, 1.14)) and continuity of care (eg, in major cities 1.14 (1.07, 1.20)) compared with their high education-level counterparts, but less likely to have a long consultation (eg, inner regional 0.90 (0.87, 0.95)), accounting for area variation in these outcomes (online supplemental table 6). Patterns of association were found whether in major cities or more remote locations.

On the other hand, people of low education level (for a given level of need) were less likely to have a specialist service compared with their higher education counterparts, accounting for area variation (figure 2, online supplemental table 7).

Sensitivity analyses did not differ materially from the main findings.

## DISCUSSION

This study has shown that where people live (at the local area level) matters for the GP and specialist services they receive, independent of their personal characteristics. This was the case across all remoteness categories— major cities, regional and more remote areas—in NSW, Australia. Further, having accounted for where people

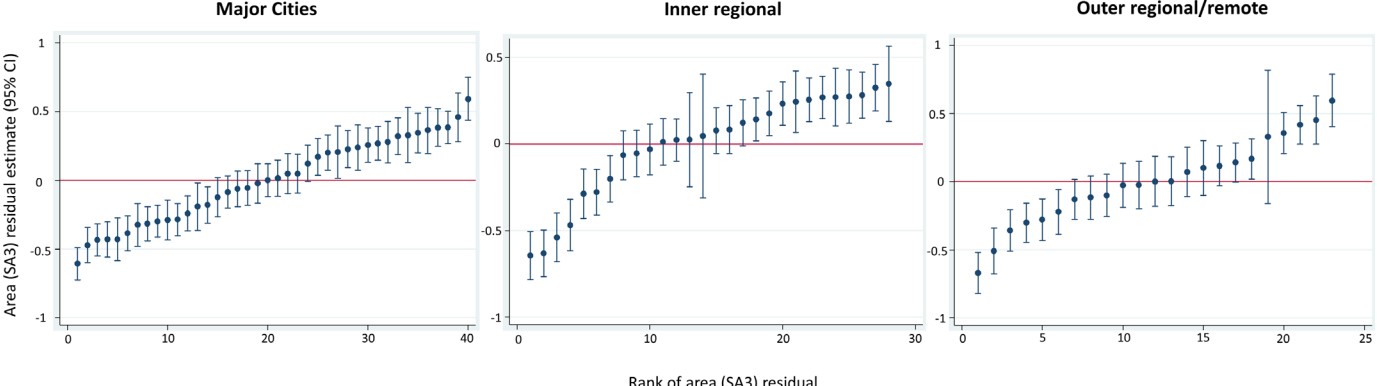

**Figure 1** Difference between mean for each area (SA3) and the mean across all areas in log odds of above-average use (95% CI) for each area, by remoteness. Adjusted for education, age, sex, country of birth, marital status, self-rated health, chronic disease and physical functional limitation; mean log odds of above-average use across areas for that remoteness category set at 0 and given by the horizontal red line; each dot represents the mean for each SA3 of the difference in log odds of above-average use for each person in that SA3 from the mean log odds of above-average use for all areas (ie, the mean of the residuals by SA3). Bars are the 95% CIs around the mean for each SA3. SA3 values that lie above and below the red with CIs that do not cross the red line are significantly different from the mean log odds of above-average use for all SA3s in that remoteness category. A person living in an area above the line has a higher probability of above-average general practitioner (GP) use than the overall sample mean, irrespective of their individual characteristics. SA3, Statistical Area Level 3.

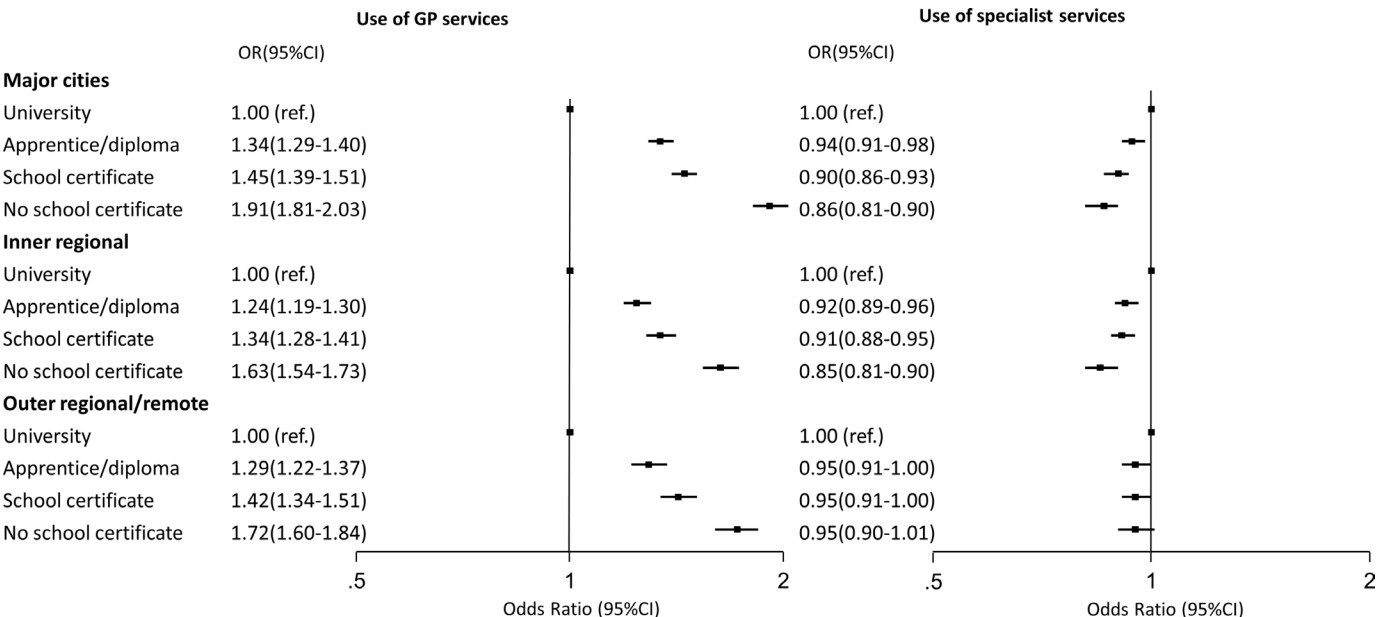

**Figure 2** ORs and 95% CIs for education with above-average use of general practitioner (GP) services and any use of specialist services, by remoteness. Model fitted with a random intercept (area level) adjusted for sociodemographic (education, age, sex, country of birth, marital status) and need (self-rated health status, number of chronic diseases, physical functioning) variables. GP use, Wald joint test of significance for education, p<0.001 for all remoteness categories. Any specialist use, Wald joint test of significance for education, cities and inner regional <0.001, outer regional not significant.

live, use of GP services and quality of care was equitable, in that, for a given level of need, disadvantaged people were more likely to use more services on average and to have continuity of care and care planning. However, the finding that advantaged people were more likely to see a specialist or have a long consultation suggests a potential source of inequity.

This is the first study in Australia, and one of few internationally, to quantify area-level variation in GP and specialist use, independent of the characteristics of people who lived in these areas. The amount of variation between areas quantified in this study is comparable to that previously reported when examining other healthcare outcomes in Australia (eg, hospitalisations[29]) and internationally.[30] More use of GP services and care planning and greater continuity of care among people of lower SEP has been previously shown in Australia[7 8 11 13 31 32] and internationally.[10 33 34] This study confirms that this holds having accounted for where people live. Previous studies have demonstrated inequity of specialist use.[10 11] However, only one study accounted for where people live, and this was in a country without gatekeeping mechanisms in place where a person can consult a specialist without referral from a GP or primary care physician.[30] Our study further confirms this finding in a context of gatekeeping policies.

We found that individual use and quality of GP and specialist services varied across small areas, for all remoteness categories, beyond what could be explained by the characteristics of people living in those areas. This suggests that there are aspects within peoples' local context that systematically shape the care of all who live in that area. The specific reasons are unknown but may relate to how services are organised and delivered (including availability of providers) within an area or structural policies determining the geographical distribution of services and providers. International multilevel studies in countries with[35] and without[30] a gatekeeping mechanism have shown that availability of GPs and specialists within an area was associated with specialist use. This has not been investigated for GP service use or quality of care. Importantly, how services are organised can be changed (through policy and practice) and doing so may contribute to reducing the unwarranted variation across areas.

Our finding of more care planning and greater continuity of care among people with lower levels of education highlights an important source of equity in the Australian primary healthcare system. People of lower SEP are more likely to have multiple and complex health and psychosocial care needs[24 36] than their advantaged counterparts, and continuity of care and care planning are essential for enabling these needs to be met. In Australia, practices and/or clinicians may provide Medicare-subsided services without the patient making an additional copayment (referred to as bulk billed). A possible explanation for the association with continuity of care observed here is that disadvantaged people are more likely to attend such services, thereby limiting the number of providers available to them. Future research exploring this will be informative for understanding the mechanism underlying this relationship.

There are likely multiple reasons why socioeconomically disadvantaged people use less specialist services for a given level of need. In Australia, GPs are incentivised

to bulk bill specific population groups. However, these are unavailable for specialists. Out-of-pocket costs for specialist services doubled in the decade prior to the study period[37] and have continued to rise since. Further, private health insurance has been shown to contribute to pro-high income use of specialist services[8 12]; yet government-funded rebates for private health insurance have remained in place. Other possible reasons include: differences in propensity to seek care due to differences in health literacy, attitudes and beliefs; or, due to negatively biased behaviours from providers, disadvantaged people are less likely to seek specialist care.[38] However, if this was the case a similar finding would be expected with use of GP services. Further, studies examining propensity to seek care[39] or rates of completion of specialist referrals[40] have not found differences between socioeconomic groups.

Alternatively, these differences may be due to provider preferences and bias. International evidence also suggests providers offer fewer services to those of low SEP[38] and are more likely to refer higher SEP individuals to a specialist.[41] Irrespective of the reasons, differences in use do not reflect need for care and hence are inequitable and unjust.

The reason why people of high education were more likely to have a long consultation is unknown. Possibly, people with lower educational attainment are more likely to be bulk billed, and given current financing arrangements in Australia, the benefit per minute falls with longer consultations. These findings may also reflect differences in health literacy. People with higher levels of education may be more likely to anticipate and expect a range of health issues to be addressed in a single episode and request a consultation length to that effect, or actively seek out practitioners with characteristics associated with longer consultations.[42]

A strength of our study is the multilevel analytical design, which allowed modelling of nested levels of data and quantification of area-level and individual-level variation. Further, the large sample linked to MBS service use allowed quantification of observed use (rather than self-report) after accounting for a range of factors. While MBS data will capture nearly all GP services, there are some settings where services provided do not attract an MBS claim. For example, publicly funded community health centres and some GP services provided in emergency departments in rural and remote areas. In addition, a substantial proportion of specialist services in Australia are provided in publicly funded hospital-based outpatient clinics, which generally do not attract an MBS rebate. Low-SEP people are more likely to use these community and hospital-based services,[8] and exclusion of these services may bias estimates for SEP gradients to be pro-high SEP. However, previous studies found this did not alter estimates of socioeconomic variation in GP and ambulatory specialist care.[8] The 45 and Up Study is not representative of the NSW population,[43] and while representativeness is not necessary for internal validity (ie,

relative effect estimates),[43] patterns of association may differ for other age groups or in other settings.

## Implications for policy and practice

An effective primary care system requires ready and reliable access to secondary-level care. This has not been equitably achieved in Australia—despite the presence of universal health insurance—undermining the equity that exists in the primary care system. National structural policies, such as minimising out-of-pocket costs (for example, through bulk billing incentives), would go some way to redressing inequitable use of specialist services. Given that private health insurance contributes to pro-high SEP use of specialist services,[13] offsetting government rebates in favour of lower income or disadvantaged individuals could also contribute to reducing this inequity. It could be argued that the inequity in community-based specialist services is balanced by a pro-low SEP preference for specialist outpatient services through the public hospital sector. However, waiting times for less urgent and more discretionary health needs (and in some instances for more urgent health needs) in the public sector are understood to exceed that in the private sector,[44] although actual wait times are not published. This increases the impact of illness on recovery and quality of life, affecting those who are disadvantaged to a greater extent. As such, addressing inequalities in access to specialist care is even more pressing.

The unwarranted variation in both GP and specialist use at the area level suggests that additional policy approaches are needed that are directed to local contexts, in addition to individuals. For example, it may be that availability of providers (both GPs and specialists) may need to be addressed, as international studies have shown this explains some of the area-level variation in care. Similarly, there may be other aspects of how services are organised and delivered at the local area level that may determine peoples' use of services. The specific drivers, and hence policy solutions to addressing the unwarranted area-level variation, require further exploration.

## CONCLUSION

It is reassuring that, for a given level of need, GP service use and important aspects of quality of care (such as care planning, continuity of care) favour those who are disadvantaged; further, this is the case regardless of where people live. However, the ongoing pro-high SEP use of specialist service threatens to undermine this and requires urgent attention. Equity measures to improve affordability are an important avenue to address this. However, both GP and specialist care varies between major cities and more remote locations, and within at the small area level. This is unwarranted and highlights an important opportunity to improve equity in the Australian healthcare system.

**Acknowledgements** This research was completed using data collected through the 45 and Up Study (www.saxinstitute.org.au). The 45 and Up Study is managed

by the Sax Institute in collaboration with the major partner, Cancer Council NSW, and the following partners: the Heart Foundation, NSW Ministry of Health, NSW Department of Communities and Justice, and Australian Red Cross Lifeblood. We thank the many thousands of people participating in the 45 and Up Study.

**Contributors** DCB, LJ, SL and RJK conceived and designed the analysis. DCB completed the data analysis, drafted the manuscript and is guarantor. All authors revised the work for intellectual content and approved the final version of the manuscript.

**Funding** This research was supported through a grant from the Australian Government through the National Health and Medical Research Council Postgraduate Scholarship (GNT1038903).

**Competing interests** None declared.

**Patient and public involvement** Patients and/or the public were not involved in the design, or conduct, or reporting, or dissemination plans of this research.

**Patient consent for publication** Not applicable.

**Ethics approval** Ethics approval for this project was obtained from the NSW Population and Health Services Research Ethics Committee (HREC/13/CIPHS/8), the University of Western Sydney Ethics Committee (H9835) and the Australian National University Human Research Ethics Committee (2011/703). Ethics approval for the 45 and Up Study was granted by the University of New South Wales Human Research Ethics Committee. The 45 and Up Study participants consented to data linkage at baseline. Linkage of the MBS data is performed under approvals from the ethics committees of Services Australia and the Australian Government Department of Health.

**Provenance and peer review** Not commissioned; externally peer reviewed.

**Data availability statement** Data may be obtained from a third party and are not publicly available. The data that support the findings of this study are available from the Sax Institute, NSW, but restrictions apply to the availability of these data, which were used under licence for the current study, and so are not publicly available. Data part of the Sax Institute's 45 and Up Study is available for approved projects to approved researchers (www.saxinstitute.org.au).

**ORCID iDs**
Danielle C Butler http://orcid.org/0000-0003-4870-4544
Sarah Larkins http://orcid.org/0000-0002-7561-3202

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
