## [Reviewer comments · BMJ Open]

ARTICLE DETAILS

TITLE (PROVISIONAL)	Does use of GP and specialist services vary across areas and according to individual-socioeconomic position? A multilevel analysis using linked data in Australia.
AUTHORS	Butler, Danielle; Larkins, Sarah Editorial Board Member; Jorm, Louisa; Korda, Rosemary

VERSION 1 – REVIEW

REVIEWER	Wändell, Per Karolinska Institutet, Neurobiology, Care Sciences and Society
REVIEW RETURNED	04-May-2023

GENERAL COMMENTS	This is an interesting manuscript on the association between socio-economic status (SES) both on area and individual level and use of GP and other specialist medical services. The study is well performed and well described. Comments: 1. The authors mention that the response rate of the 45 and Up Study has a response rate of 18%. Could this have affected the results in any way?2. Earlier studies have shown an association between low SES and morbidity, and thus higher care needs. Such studies could have been referred to.3. Earlier studies, even if few, have found a lower rate of specialist care among individuals with lower SES. Such studies could also be referred to.
---

REVIEWER	Carter, Mary University of Exeter, Health & Community Sciences
REVIEW RETURNED	22-May-2023

GENERAL COMMENTS	This is a well-written paper concerning an important topic. Unfortunately, my understanding of statistics is not sophisticated, so I cannot comment extensively on the statistical methods used - using odds ratios seems reasonable as they are relatively straightforward to interpret for a range of skill levels. I have indicated that specialist statistical review may be necessary (above). I have also noticed that almost half of the references (17 out of 40) date from 2010 or earlier - the authors may wish to consider including some more up-to-date references.
--

	Reference no. 37 (Sorensen, Olsen et al) does not include a year - apologies for the picky point.
--	---

REVIEWER	Doust, Jenny The University of Queensland Faculty of Medicine and Biomedical Sciences
REVIEW RETURNED	14-Sep-2023

GENERAL COMMENTS	Area-level and individual-socioeconomic variation in use of GP and specialist services This is an interesting paper with some results that are potentially useful to those who are interested in health care delivery, especially in the Australian context. The main message of the findings are important, but could be made clearer. My main suggestion for the paper is to provide sub-headings in the section under the heading “variables” for:  a) Outcomes of interest b) Exposure variables c) Covariates Which variables are the exposure, the outcome and the covariates needs to be clearer in the abstract. The abstract refers to the “individual socio-economic position” but this is measured by the educational attainment level of individuals. The design section should clearly state this. If this requires some extra words, the second sentence of the current abstract could be omitted. The term “individual characteristics” is used a few times in the abstract and the introduction, and it is not clear what variables are being included in this term. It is also not currently clear in the abstract what a mean odds ratio means, and it needs to be explained in a way that can be understood without going to the methods of the paper. The acronym PC is used in the abstract without explanation. In the conclusion, the sentence “Specialist care but not GP care was inequitable” would be better worded as “Access to GP care was equitable, but specialist care was not. In the 45 and up study, what proportion of the total NSW population aged 45 and older were invited to participate in the study, or is this the 11%? I was confused by the first sentence in the analysis section “Participants were followed for one year after study entry (most had completed entry by 2008). What was the follow-up time for the participants in the study?
--

	A statistician will need to check the methods described in the second paragraph of the analysis section. In the section under individual level socioeconomic variation, I would explain the results in qualitative sentences eg there was increasing use of general practice services with decreased educational attainment, etc, and leave the numbers for the Figure. It is quite difficult to work out what the numbers mean in this paragraph as it is currently written. In the discussion section, add in a comment that access to GP services also seem to be distributed equitably according to healthcare needs as measured by self-reported health etc. This is a surprising finding. Is a reason for the fewer observed long consultations in people with lower educational attainment (and this would be a preferable term to the current “low education”), is that they are more likely to attend practices that exclusively bulk bill, as these practices tend to be in outer metropolitan areas? Table 1, the last column needs to be clearer when the numbers presented here are totals and when they are averages. Also, in the row looking at long consult yes or no, does this mean that a long consult has been billed at any time? Maybe it should be labelled “Any long consult”?
--	--

REVIEWER	Youens, David Curtin University, School of Population Health
REVIEW RETURNED	18-Sep-2023

GENERAL COMMENTS	This paper presents an investigation of use of health services according to individual socio-economic status accounting for area-level variation in use. The combination of self-reported and administrative data is a strength, the multi-level approach is useful and the paper is well written. I have only minor comments that I think will strengthen the paper. General comments: When referencing additional files in text would it be possible to reference the specific table number? Methods: Under “Study population and setting” there is mention of the 5-yearly questionnaires. As only the baseline questionnaire is used for this study, mention of the follow-up surveys might not be necessary (though if the terminology used has already been approved by the Sax Institute, please ignore this). The variables selected as outcomes make sense, and the variables used as adjustors for need make sense given the outcomes being used. Under the Analysis subheading, you state that model 1 was a random intercept model with no explanatory variables to determine if outcomes varied at the area level. But, in the introduction you state that you aim to examine the extent to which GP and specialist service use varied at the area-level, having accounted
--

	for the characteristics of people who lived in those areas. And then through most of the results and Figure 1 you focus on the area-level variation having adjusted for individual characteristics. I think that most of the investigation of area-level variation is based on Model 2, with Model 1 included in supplementary tables only without much interpretation. Could the methods text be clarified, as this led me to re-read parts multiple times. Results: If there is space, an explanation of the interpretation of MOR may be useful in either the methods or results, as some readers may be unfamiliar with this. Note that for specialist use, the abstract and supp. table 5 say MOR 1.16-1.18 while results text (Area level variation subheading) says 1.16-1.17 I believe that all analyses relating to the continuity of care outcome are restricted to a smaller cohort with a higher general level of need due to the restriction to ≥ 4 GP contacts. This should probably be made clear in table footnotes or in text, and the authors should consider whether the interpretation of these results would be impacted by this at all. Discussion: The term bulk-billing should be explained for non-Australian readers. The discussion includes less on findings regarding continuity than on other outcomes. I wonder if financial aspects may play a role here as you suggest for some other findings – i.e. those who are disadvantaged may be restricted to visiting bulk-billing GPs only, hence a lower number of providers are realistically available, and a lesser opportunity for discontinuity? This is a suggestion only, to consider including if the authors agree with this interpretation.
--	--

VERSION 1 – AUTHOR RESPONSE

Reviewer 1		
2. The authors mention that the response rate of the 45 and Up Study has a response rate of 18%. Could this have affected the results in any way?	We have addressed this under limitations as follows. Line 364 page 13	The 45 and Up Study is a representative sample of the NSW population (not necessarily representative of other age groups).
3. Earlier studies have shown an association between low SES and morbidity, and thus higher care needs. Such studies could have been referred to.	We have amended the text as follows, and included appropriate references. Line 162 page 7	SEP is associated with higher health care needs (24, 25), which may affect health care.

4. Earlier studies, even if few, have found a lower rate of specialist care among individuals with lower SES. Such studies could also been referred to.	We have referred to this in the introducon as per Introducon, paragraph 2, line 91. We have updated our references as per reviewer 2's suggeson. In discussion we make the point that there one internaonal study, to the authors' knowledge, showing inequity in specialist use having accounted for where a person lives. We have amended the text for clarity. Line 292 page 11.	On the other more likely to lowSEP. Previous stud specialist use accounted fo country witho where a pers referral from
--	---	--

		physician(3 finding in a
Reviewer 2.		
5. I have also noced that almost half of the references (17 out of 40) date from 2010 or earlier - the authors may wish to consider including some more up-to-date references. 6. Reference no. 37 (Sorensen, Olsen et al) does not include a year - apologies for the picky point.	We have updated references where appropriate throughout the manuscript. We have added the date to the reference indicated.	
Reviewer 3		
7. My main suggeson for the paper is to provide sub-headings in the secon under the heading "variables" for:  a) Outcomes of interest b) Exposure variables c) Covariates 	We have inserted sub-headings for clarity as suggested.	
8. Which variables are the exposure, the outcome and the covariates needs to be clearer in the abstract. The abstract refers to the "individual socio-economic posion" but this is measured by the educaonal atainment level of individuals. The design secon should clearly state this. If this requires some extra words, the second sentence of the current abstract could be omitted.	We have amended the text of the abstract as suggested for clarity. In the results of the abstract we have stated educaon level as our main exposure variable	Design: Our ofcare and regression v raos (MORs outcomes, v of moving to and ii) ORs outcomes a exposure va individual se characterise remoteness
9. The term "individual characteriscs" is used a few mes in the abstract and the	We have amended the text in the abstract and introducon to clarify this refers to	To date, no individual se

introducon, and it is not clear what variables are being included in this term.	sociodemographic and health characteristics. See response 7 above and as follows in the introducon, last 2 paragraphs.	primary and area-variation extent of variation explained by sociodemographic living in those The aim of data and the extent to varied at the sociodemographic people who
10. It is also not currently clear in the abstract what a mean odds ratio means, and it needs to be explained in a way that can be understood without going to the methods of the paper.	Please see response 7 above	
11. The acronym PC is used in the abstract without explanation.	Thank you for bringing this to our attention, we have amended the text accordingly.	
12. In the conclusion, the sentence "Specialist care but not GP care was inequitable" would be better worded as "Access to GP care was equitable, but specialist care was not.	We have amended the text as suggested. We have retained reference to use as more accurately reflecting what was measured.	
13. In the 45 and up study, what proportion of the total NSW population aged 45 and older were invited to participate in the study, or is this the 11%?	We have clarified this in the text as follows. Line 115 page 5	About 19% and participated population aged
14. I was confused by the first sentence in the analysis section "Participants were followed for one year after study entry (most had completed entry by 2008). What was the	We have clarified this in the text. Line 178, page 7.	Participants completing the completed the continuity of
follow-up time for the participants in the study?		
15. In the section under individual level socioeconomic variation, I would explain the results in qualitative sentences eg there was increasing use of general practice services with decreased educational attainment, etc, and leave the numbers for the Figure. It is quite difficult to work out what the numbers mean in this paragraph as it is currently written.	We have amended the text as suggested. Given odds ratios for secondary outcomes (quality of GP care) are reported in the additional file, we have left these figures within the text.	People of low services on higher level and having (Figure 2).

16. In the discussion section, add in a comment that access to GP services also seem to be distributed equitably according to healthcare needs as measured by self-reported health etc. This is a surprising finding.	We have clarified the text in the discussion where we have raised this point as follows.	Further, having use of GP services is equitable, in disadvantaged areas more services care and ca
17. Is a reason for the fewer observed long consultations in people with lower educational attainment (and this would be a preferable term to the current “low education”), is that they are more likely to attend practices that exclusively bulk bill, as these practices tend to be in outer metropolitan areas?	We agree that a possible explanation is that people with lower educational attainment may be more likely to attend practices that bulk-bill. However, our study did not examine this, and hence we are unable to make inferences in relation to this. We have noted in the discussion that individuals of low education are more likely to be bulk-billed and given the benefit per minute decreases with length of consultations this may in part explain this relationship (line 347, page 13). We are uncertain what the reviewer means with respect to outer metropolitan areas, as we have	
	demonstrated this association accounting for variation across areas in the probability of a long consult.	
18. Table 1, the last column needs to be clearer when the numbers presented here are totals and when they are averages. Also, in the row looking at long consult yes or no, does this mean that a long consult has been billed at any time? Maybe it should be labelled “Any long consult”?	We have amended the labels for the long consult outcome as suggested. We are unclear as to the query of totals vs averages as the last column of table 1 reports proportions and totals only as labelled	
Reviewer 4		
19. When referencing additional files in text would it be possible to reference the specific table number?	We have amended the text as suggested throughout the manuscript	
20. Under “Study population and setting” there is mention of the 5-yearly questionnaires. As only the baseline questionnaire is used for this study, mention of the follow-up surveys might not be necessary (though if the terminology used has already been approved by the Sax Institute, please ignore this).	This is the approved terminology required by the Sax Institute as the data custodian of the 45 and Up study	

21. Under the Analysis subheading, you state that model 1 was a random intercept model with no explanatory variables to determine if outcomes varied at the area level. But, in the introduction you state that you aim to examine the extent to which GP and specialist service use varied at the area-level, having accounted for the characteristics of people who lived in those areas. And then through most of the results and Figure 1 you focus on the area-level variation having	The intent of the random intercept model without explanatory variable was to determine the appropriateness of a random intercept model. We have amended the text to be consistent with the focus of the analysis on the fully adjusted model, and to clarify the purpose of the unadjusted model as follows.	Frequencies of the sample for the total. A two-level regression model (residence) to determine if variation in the intercept model explanatory (additional fil
adjusted for individual characteristics. I think that most of the investigation of area-level variation is based on Model 2, with Model 1 included in supplementary tables only without much interpretation. Could the methods text be clarified, as this led me to re-read parts multiple times.		tables 3-5). adjusted for and confounding individual-level outcomes (variation).
22. If there is space, an explanation of the interpretation of MOR may be useful in either the methods or results, as some readers may be unfamiliar with this.	We have amended the text in the methods to provide further explanation as follows. Line 201, page 8 We have also explained the interpretation in the results line 246 page 9 section 'Area-level variation'.	Conceptual ratios calculated for people with different area risk that would be another with This means with a higher (median) 32 above-average identical characteristics lower rate of
23. Note that for specialist use, the abstract and supplementary table 5 say MOR 1.16-1.18 while results text (Area level variation subheading) says 1.16-1.17	Thank you for bringing this error to our attention. We have amended the text accordingly.	
24. I believe that all analyses relating to the continuity of care outcome are restricted to a smaller cohort with a higher general level of need due to the restriction to ≥ 4 GP contacts. This should probably be made clear in table footnotes or in text, and the authors should consider whether the interpretation	We have amended the text in the methods to clarify this as follows.	Participants completing the completed the continuity-of-care had a least end of the first identifier code continuity of

of these results would be impacted by this at all.	And in the footnotes of table 1. It is possible that people with infrequent use of GP services may also have good continuity of care, as measured by self-reported relational continuity of care. However, such measures were unavailable in these data. It is uncertain whether this may have altered the results. We believe that further discussion of this would detract from the main message of the paper.	in the follow up period, and for analyses of care planning, had at least one long-term health condition. Notes: N, number; %, percentage; Columns for each variable category for each educational attainment categories sum to 100%. Values in last column gives break down by category for each individual variables for the total sample, not stratified by educational attainment. For each variable, total (n) sums to 265,083 and percent sums to 100% including missing data; Chi-squared test for trend with education $p < .001$ all variables. Missing: age < 1%, country of birth 1%, marital status 0.6%, self-rated health 3.5%, physical functioning 13.3%. For analyses of continuity of care, those who died in the second year or had less than four GP MBS services during follow-up were excluded (n=31595). Additional exclusions included: i) care planning participants without a chronic disease (n=81222); and ii) outlier observations of long consultations (n=3).
25. The term bulk-billing should be explained for non-Australian readers.	To address this and in response to your following suggestion we have amended the text as follows. Line 319, page 12	Our finding of more care planning and greater continuity of care among people with lower levels of education highlights an important source of equity in the Australian primary healthcare system. People of lower SEP are more likely to have multiple and complex health and psychosocial care needs (24, 36) than their advantaged counterparts, and continuity of care and care planning are essential for enabling these

	And for clarity we have amended the following text that follows in the next paragraph of the paper. Line 330 page 12	needs to be met. In Australia, practices and/or clinicians may provide Medicare subsidised services without the patient making an additional co-payment (referred to as bulk-billed). A possible explanation for the association with continuity of care observed here, is that disadvantaged people are more likely to attend such services, thereby limiting the number of providers available to them. Future research exploring this will be informative for the underlying mechanism for this relationship. In Australia, GPs are incentivised to bulk-bill specific population groups. However, these are unavailable for specialists.
26. The discussion includes less on findings regarding continuity than on other outcomes. I wonder if financial aspects may play a role here as you suggest for some other findings – i.e. those who are disadvantaged may be restricted to visiting bulk-billing GPs only, hence a lower number of providers are realistically available, and a lesser opportunity for discontinuity? This is a suggestion only, to consider including if the authors agree with this interpretation.	We agree that a preference/necessity to attend bulk-billing GPs, hence limiting the number of possible providers is a possible explanation for the observed association. Examining this was beyond the scope of the current paper. We have amended the text to highlight this possible explanation as per above	

VERSION 2 – REVIEW

REVIEWER	Doust, Jenny The University of Queensland Faculty of Medicine and Biomedical Sciences
REVIEW RETURNED	30-Nov-2023

GENERAL COMMENTS	All comments from myself and other reviewers have been appropriately addressed by the manuscript authors. I look forward to seeing the article in print.
--

REVIEWER	Youens, David Curtin University, School of Population Health
REVIEW RETURNED	13-Nov-2023

GENERAL COMMENTS	The authors have addressed all of my comments from the original submission. I think this is a well written paper and enjoyed reading it.
--